# Snapshot of the Probiotic Potential of *Kluveromyces marxianus* DMKU-1042 Using a Comparative Probiogenomics Approach

**DOI:** 10.3390/foods12234329

**Published:** 2023-11-30

**Authors:** Mati Ullah, Muhammad Rizwan, Ali Raza, Yutong Xia, Jianda Han, Yi Ma, Huayou Chen

**Affiliations:** 1School of Life Sciences, Jiangsu University, Zhenjiang 212013, China; matiesh@ujs.edu.cn (M.U.); 2222217002@stmail.ujs.edu.cn (Y.X.); 2222217014@stmail.ujs.edu.cn (J.H.); mayiayy@126.com (Y.M.); 2College of Fisheries, Huazhong Agriculture University, Wuhan 430070, China; rizwiktk007@hotmail.com; 3School of Life Sciences, University of Science and Technology of China, Hefei 230026, China; ali512@mail.ustc.edu.cn

**Keywords:** *Kluyveromyces marxianus*, yeast, probiotic, next-generation sequencing, fermentation, genes

## Abstract

*Kluyveromyces marxianus* is a rapidly growing thermotolerant yeast that secretes a variety of lytic enzymes, utilizes different sugars, and produces ethanol. The probiotic potential of this yeast has not been well explored. To evaluate its probiotic potential, the yeast strain *Kluyveromyces marxianus* DMKU3-1042 was analyzed using next-generation sequencing technology. Analysis of the genomes showed that the yeast isolates had a GC content of 40.10–40.59%. The isolates had many genes related to glycerol and mannose metabolism, as well as genes for acetoin and butanediol metabolism, acetolactate synthase subunits, and lactic acid fermentation. The strain isolates were also found to possess genes for the synthesis of different vitamins and Coenzyme A. Genes related to heat and hyperosmotic shock tolerance, as well as protection against reactive oxygen species were also found. Additionally, the isolates contained genes for the synthesis of lysine, threonine, methionine, and cysteine, as well as genes with anticoagulation and anti-inflammatory properties. Based on our analysis, we concluded that the strain DMKU3-1042 possesses probiotic properties that make it suitable for use in food and feed supplementation.

## 1. Introduction

Microorganisms exhibit great potential as a prospective reservoir for the generation of value-added products and metabolites. A cell functions synonymously to a highly synchronized factory to efficiently synthesize a myriad of important secondary materials, metabolites, enzymes, etc. [1]. A multitude of microorganisms, mainly bacteria such as *Lactobacillus rhamnosus*, *Bifidobacterium longum*, *Bifidobacterium bifidum*, and *Bifidobacterium animalis*, along with the yeast *Saccharomyces cerevisiae* are reported with enormous potential of biomolecules production [2,3,4]. In yeast, *Saccharomyces cerevisiae* is the most used strain for producing targeted products [5]. The reason is its known genomic profile, easy and safe engineering of its genetic DNA, and high product yield [6]. But other natural yeasts hold even more vital characteristics for commercial applications and can offer some benefits over *S. cerevisiae* [7]. We make the presumption that such yeasts may have the capability to become a model microbial cell factory. Recently, *Kluyveromyces marxianus* has been reviewed as an alternative yeast strain [8]. This yeast strain is homothallic, haploid, hemi-ascomycetous, and thermotolerant and is naturally available in fruits and fermented dairy products [9]. *K. marxianus* has more similar features to that of *Kluyveromyces lactis* [10]. Both yeasts are capable of lactose metabolism, lacking in *Saccharomyces cerevisiae.* The *K. marxianus* is a source of several biomolecules, mainly enzymes, i.e., acid phosphatase, protein phosphatases, carboxypeptidases, transporters, and assimilators of sugars (glucose, fructose, lactose, galactose, xylose, and organic acids like malic acid and lactic acid) [11]. Additionally, it contain genes for metabolism of inulinase, β-glucosides, β-galactosids, and endopolygalacturons [12]. *K. marxianus* possess metabolic diversity, which opens up a route to analyze the physiology and metabolism of this yeast for several industrial potential applications [10]. Thus, *K. marxianus* has several advantages, i.e., fermentation of multiple sugars (glucose, lactose, xylose, and arabinose), can perform fermentation up to 52 °C temperature [13], absence of glucose inhibition, and inulin fermentation [13]. Importantly, *K. marxianus* has the fastest growth rate among all the reported yeast strains known so far [8]. It has been used in the production of industrial enzymes, heterologous proteins and bio-ingredients, single-cell proteins, bioethanol production, and also has bioremediation applications [13]. *K. marxianus* produces an extracellular endopolygalacturonase (EPG) and other hydrolases to degrade agriculture biomass, which makes it a feasible choice for bioprocessing technological applications. Genetic engineering tools have been developed for non-homologous end-joining activity in *K. marxianus* [14]. To achieve a better genetic insight into *K. marxianus,* genome sequences of *K. marxianus* can facilitate it to be a highly proficient yeast for future developments [15]. Their properties of rational genetic manipulation, metabolic engineering, culture growth, and fermentation process yield/productivity have resulted in a host capable of synthesizing a heterologous compound, and have resulted in generally-regarded-as-safe (GRAS) status being given to *K. marxianus* strains [10]. *K. marxianus* possesses all the capabilities to be a model organism in industrial applications. *K. marxianus* has been reported to use the thermostable endo-β-1,4-glucanase, β-glucosidase, and cellobiohydrolase for hydrolyzing cellulosic biomass into ethanol [16].

Probiotics are defined as “live microbes, when administered in adequate amounts confer health benefits onto the host”. Numerous health-associated benefits have been associated with probiotics, including, but not limited to, aiding digestion, preventing infections and diseases, and improving the immune system [17]. The largest proportion of currently used probiotics belong to lactic acid producing bacteria, especially members of the genera *lactobacillus* and *bifidobacterium* [18]. In addition, some non-lactic acid producing bacteria, such as *bacillus* and *propionibacterium*, are also recommended as probiotics [19]. Along with bacterial probiotics, some nonpathogenic yeasts such as *Saccharomyces cerevisiae, Saccharomyces boulardii,* and *Kluyveromyces marxianus* are also used as probiotics [20]. *K. marxianus* is a strong candidate as a probiotic of choice for food and the food industry [21]. Furthermore, this yeast has been placed in the category of “generally recognized as safe” (GRAS) by the Food and Drug Administration (FDA) and “qualified presumption of safety” (QPS) by the European Food Safety Authority (EFSA) [22].

The probiotic attributes of microorganisms are specie and strain specific. This notion has been supported where the efficacy of different probiotics has been evaluated [23]. It was found that only 4 out of 127 probiotic *Lactobacillus* strains were able to bear bile salts and stomach acidity [24]. Similarly, an evaluation of more than 170 *Lactobacillus* species showed variation in probiotic-associated characteristics [25]. Furthermore, the strain *Saccharomyces boulardii* CNCM I-7145 was found to have multiple antipathogenic properties [26]. Therefore, the international probiotic guidelines and recognized experts recommended strain-specific descriptions when reporting their results [27]. Physiological traits, such as metabolic profile, enzyme and vitamin production, antibiotic resistance, and antipathogenic activity, are determined by the genetic material of probiotics [28]. The study of the DNA of a probiotic provides insight into its physiological properties, potential probiotic attributes, and safety concerns. The Food and Drug Administration (FDA) and World Health Organization (WHO) also encourage the genomic study of newly isolated probiotics [29]. Therefore, the present investigation sought to explore the probiotic capabilities of *K. marxianus* DMKU3-1042 through the utilization of cutting-edge sequencing and computational methodologies. Furthermore, through exploration of specific attributes, the genomes were compared with the genomes of other available strains through pangenome analysis.

## 2. Materials and Methods

### 2.1. Yeast Strain, Growth Media, and Culture Condition

The yeast strain used in the current study, *K. marxianus* DMKU 3-1042, was isolated from traditional fermented yogurt samples. A total of five different traditional fermented yogurt samples (made from cow milk) were collected from various local vendors in the Karak district of Khyber Pakhtoonkhwa, Pakistan. The samples were serially diluted in phosphate-buffered saline (PBS). The different dilutions of each sample were plated on yeast extract peptone dextrose (YPD) agar medium (Jinan Babio Biotechnology Co., Ltd., Shandong, Jinan, China) comprising 20 g of peptone, 10 g of yeast extract, and 20 g of glucose per liter of distilled water and incubated for a period of 48–72 h at a temperature of 30 °C. Yeast cultures were routinely cultivated in YPD broth (1% yeast extract, 2% peptone, 2% glucose) at a temperature of 30 °C and pH 6.4 for a period of 24–48 h.

### 2.2. Genomic DNA Isolation and Library Preparation

The genomic DNA of yeast was extracted utilizing the FavorPrep Fungi/Yeast Genomic DNA Extraction Mini Kit (Favorgen, Taiwan) in strict accordance with the manufacturer’s instructions. The quantification of the isolated DNA was conducted using the Qubit 2.0 fluorometer (Invitrogen, Carlsbad, CA, USA). The DNA libraries were prepared using the Vazyme TruePrep™ DNA Library Prep V2 Kit for Illumina (Vazyme Biotech, Nanjing, China). Sequencing was performed on Illumina HiSeq-2000 sequencing platform (BGI-Shenzhen, Shenzhen, China) according to instructions provided by the manufacturer.

### 2.3. Data for Comparative Analysis

Apart from the five isolates of *Kluyveromyces marxianus* DMKU3-1042 sequenced in this study, we also selected freely available samples of sequenced yeast genomes isolated from different sources for the comparative study. The detail information of the analyzed samples is listed in Appendix A. All datasets were obtained from the National Center for Biotechnology Information (NCBI) Sequence Read Archive (SRA) repository.

### 2.4. Genome Assembly and Annotation

The isolates of *Kluyveromyces marxianus* DMKU3-1042 strain were subjected to reads processing and cutadapt (v.2.2) was used to trim adapter sequences. For Genome assembly, we used unicycler v0.4.8 assembler followed by a two-polishing round of Pilon 1.23 used to improve draft genome assemblies and the quality of assemblies was evaluated using QUAST (v 5.0.2) [30]. 15 *K. marxianus* genome strains (NRRL Y-6860, Olga-1, Olga-2, SLP1000000F_001, UFS-Y2791, KM3, L03, LHW-O, NBRC 1777, DMKU3-1042, CBS6556, FIM1, IIPE453, and DMB1) were downloaded from NCBI genome database for comparative study which includes functional genome features, phylogenomics, average nucleotide identity (ANI), and pangenome analysis. Genome annotation of the five isolates of *K. marxianus* DMKU3-1042 genomes and other 15 strains downloaded from the NCBI database was performed using companion fungi annotation pipeline [31].

### 2.5. Phylogenetic and Average Nucleotide Identity (ANI) Analysis

A whole-genome-based phylogenetic tree was constructed using the branch support method (RAxML Fast Bootstrapping) [32] on BV-BRC online tool which uses PGFams (global (cross-genus) protein families) as homology groups [33]. During the phylogenetic analysis, 1000 single-copy genes were requested for alignment. The MAFFT alignment program was used to align 1000 proteins and coding DNA found to be used for RAxML analysis. We used MEGA v11.0 for the tree visualization [34]. The JSpecies was used to calculate the average percentage nucleotide identity (ANI) between the whole genomes [35].

### 2.6. Pangenome Analysis

For whole-genome comparison, pangenome analysis was performed which consists of a comparison of chromosomal properties of the *K. marxianus* genome strains, and the genome map (circular view) was generated by using BRIGS 0.95 software to show the circular genome map [36].

## 3. Results and Discussion

### 3.1. Comparative Genomic Analyses

The comparative genomic analysis is aimed at investigating specific functional gene categories with probiotic potentials of the isolates, likewise the phylogenetic correlation, identification of the entire set of strain-specific genes via pangenome analysis, and the niche association grouped in terms of geographical location and isolation source. Hence 15 publicly available *K. marxianus* genomes strains downloaded from the NCBI database with geographical localization and isolation sources were, respectively, displayed as follows: Japan (*n* = 1)/Sugarcane, India (*n* = 1)/Sugar mill waste, South Africa (*n* = 1)/Agave plant, Ecuador (*n* = 1/Banana alcoholic beverage, South Korea (*n* = 1) Onion, Germany (*n* = 2)/Kefir, China (*n* = 2)/Tibetan Kefir and yogurt, and Mexico (*n* = 2)/Fermented maize milk, fermented tanks and Pozol including others indicated as—(not available) as shown in Appendix A. Comparative genomics is a technique employed to assess multiple genomes of probiotic bacterial strains. It is referred to as pan-probiosis [37]. This approach enables the identification of genes associated with probiotic properties. These genes can either be shared among strains or unique to a particular bacterium genus or species. Additionally, when integrated with phylogenomic analyses, this strategy allows for the correlation of genotypes and phenotypes. This correlation aids in the selection of strains for specific clinical or biotechnological applications [38].

### 3.2. Genome Information/Chromosomal Properties

The chromosomal properties for the five isolates of *K. marxianus* DMKU3-1042 genome sequences ranges in size from 9.80 to 9.88 Mb and the annotation process for each isolate as shown in Table 1, reported S1, S2, S3, S4, and S5 isolate annotation to predict 9875, 9870, 9870, 9875, and 9870 total coding sequences (CDSs). The gene density per samples (S1 = 71.25%, S2 = 71.60%, S3 = 71.20%, 4 = 71.10%, S5 = 71.30%.) and their GC contents were reported to be 41.69%, 41.10%, 41.60%, 41.25%, and 41.90%, respectively. The chromosomal properties of all five isolates to the collected genomes from the NCBI database showed (Table 1) close relativity when compared to each other. The estimation of genome size is a crucial task in genome surveys. Determining the genome size is of significant importance for selecting sequencing strategies, sequencing depth, and evaluating assembly outcomes. Genome size and average gene density analysis is also important for understanding the ecological and evolutionary forces acting on microorganisms within an environment. From an ecological perspective, microbial genome size may reflect environmental complexity, metabolic lifestyle, and community niche [39]. GC content evaluation is also an important component of microbial genome surveys. It involves plotting the GC content and coverage distribution of preliminary assembled contigs [40].

### 3.3. ANI, Phylogenetic Analyses and Whole-Genome Comparison of Kluyveromyces marxianus DMKU3-1042 Strain

The constructed phylogenetic tree showed four clusters from the root based on comparative tree analysis at the whole-genome level (Figure 1), different colors of lines were each signed representing the clades (red represented Clade A, Blue represented Clade B, Green represented Clade C, and Black Represented Clade D). All strains in the first clade (composed of IIPE453, DMKU3-1042 GCF, FIM1, DMB1, and DMB0901) were isolated in Europe except KCTC1755 which was isolated in America. In the second clade, the strains were isolated in different locations (Mexico, Japan, and South Africa) and were composed of CBS6556, NBRC 1777, and LHW-0. In the third clade (composed of NRRL Y-6860, Olga 1, L03, DMKU3-1042 S4, Olga 2, DMKU3-1042 S3, DMKU3-1042 S1, DMKU3-1042 S5, DMKU3-1042 S2, and KM3), all five isolates of *K. marxianus* DMKU3-1042 genome sequences were grouped with other strains in Ecuador, Germany, Ireland, and China. The fourth clade was composed of UFS-Y2791 isolated in South Africa and SLP1_000000F_001 in Mexico. The genome map shows the whole-genome comparison of each of the five isolates compared to other genome strains as shown in Figure 2. Phylogenetic analysis plays a crucial role in investigating diverse biological inquiries, including the interconnections between species or gene and the alterations in population dynamics and species migration [41]. Additionally, Table 2 describes the average nucleotide identity (ANI) of *K. marxianus* as compared to other genomes at the cut off 95% species demarcation criteria. The estimation of genetic relatedness between genomes is a crucial task in determining species boundaries. In recent years, the whole-genome average nucleotide identity (ANI) has emerged as a reliable method for this purpose. Typically, organisms belonging to the same species exhibit an ANI of ≥95% among themselves. ANI measures the average nucleotide identity of all orthologous genes shared between two genomes and provides a robust distinction between strains of the same or closely related species (i.e., exhibiting 80–100% ANI). It is important to note that ANI does not solely represent core genome evolutionary relatedness, as orthologous genes can vary significantly between genome pairs [42]

### 3.4. Functional Categories Related to Probiotic Capabilities

Investigating specific functional categories with genes that code for potential probiotic capabilities at the genome level is considered to be very essential in probiotic organisms, especially in comparative strains [43]. Figure 3 shows the relative distribution of genes coding for functional categories relating to probiotic capabilities in metabolism, carbohydrates, protein synthesis, energy, fermentation, etc., compared between the 5 *K. marxianus* DMKU3-1042 isolates and other strains. Among the probiotic encoding genes detected in *K. marxianus* genomes were those related to metabolism, protein fate, translation, energy, central metabolism, fermentation, respiration, RNA processing, and modification, likewise transcription, cell rescue defense, cell cycle, and DNA processing. The encoding genes related to membrane transport, cellular transport, and the transport mechanism of the functional categories were found to be common to the strains. In this study, we deeply evaluated and provided insights into the specific probiotic capabilities relative to metabolism, protein synthesis, energy, stress response, defense, and virulence. This includes differential genes coding for carbohydrate utilization (galactose, lactose, mannose, sucrose, fructose, xylose, arabinose, glycerol, and cellobiose) and fermentation, vitamins, coenzymes, prosthetic groups, stress response and defense mechanisms, amino acids, antioxidants, anticoagulants, and anti-inflammatory processes; each of these was compared between the *K. marxianus* genome strains in terms of gene contents and was found to vary between the five sequenced isolates and other investigated strains from different geographical localization and source of isolation.

### 3.5. Carbohydrate Utilization and Fermentation in Kluyveromyces marxianus Strains (Genes Relevant for Sugar Utilization)

In various commercial applications, *K. marxianus* strains were proved to be an important yeast for industrial usage due to their capacity to utilize a wide range of sugars and produce ethanol by fermentation alongside other beneficial properties [44]. In respect to this, performing annotation of *K. marxianus* genome strains, from the functional categories in metabolism, we found relevant genes that code for the utilization of certain sugars at their initial catabolism including galactose, lactose, mannose, sucrose, fructose, xylose, arabinose, glycerol, and cellobiose. As shown in Figure 4, the genes encoding each of the utilized sugar are distributed differently across the *K. marxianus* strains. Comparatively across the strains, we found that there were more genes that code for glycerol and mannose utilization than other sugars, with galactose and arabinose being the least. Specifically, in the five isolates of *K. marxianus* DMKU3-1042, both RAG2 (Galactose-1-phosphate uridylyltransferase, EC 2.7.7.10) and HXK2 (RAG5) (Hexokinase, EC 2.7.1.1) are the encoding genes relevant to glucose, fructose, and mannose utilization. PMI40 (Mannose 6-phosphateisomerase, EC 5.3.1.8) is another gene also found to encode only mannose. GAL1 (Galactokinase), GAL7 (Galactose-1-phosphate uridylyltransferase, EC 2.7.7.10), and GAL5 (PGM2) (Phosphoglucomutase, EC 5.4.2.2) encode galactose utilization while XYL1 (Xylose reductase EC 1.1.1), XYL 2 (Xylitol dehydrogenase, EC 1.1.1.9), and XKS1 (Xylulokinase, EC 2.7.1.17) encode xylose utilization. ARA1 (Arabinose dehydrogenase [NADP+ dependent]), PRS5 (Ribose-phosphate pyrophosphokinase 5), DLD1 (D-Lactate dehydrogenase [cytochrome] 1, mitochondrial), and DLD2 D-Lactate dehydrogenase [cytochrome] 2, mitochondrial encode Arabinose sugar utilization. GPD1 (Glycerol-3-phosphate dehydrogenase [NAD(+)] 1, EC 1.1.1.94), GUT2 (Glycerol-3-phosphate dehydrogenase, mitochondrial), and GUT1 (Glycerol kinase, EC 2.7.130) encode glycerol sugar utilization and are mostly distributed across isolates. GAL80 (Galactose/lactose metabolism regulatory protein) encodes lactose and β-glucosidase (EC 3.2.1.21). ASF2 (Endo-1,3(4)- β -glucanase 2, EC 2.3.1.6) encodes cellobiose sugar utilization and lastly ADH1 (Alcohol dehydrogenase 1, EC 1.1.1.1), encodes sucrose, raffinose, and inulin sugar utilization. Some of these encoded genes are widely distributed differently across the genomes while others are distributed evenly as shown in Figure 4.

### 3.6. Fermentation

The complete genome sequences of *K. marxianus* DMKU3-1042 isolates and other investigated strains were comparatively determined, which revealed many genes for the cells to carry out fermentation at elevated temperatures, making it suitable for producing cellulose ethanol and assimilate a wide variety of sugars including xylose and arabinose [8]. In our analysis, found in the subsystem functional categorization as shown in Figure 3, we found three subsystems (acetoin, butanediol metabolism, acetolactate synthase subunits, and lactic acid fermentation) with variables of genes that encode functional proteins responsible for fermentation across all strains. The proteins that are responsible for acetoin, butanediol metabolism include 2, 3-butanediol dehydrogenase, R-alcohol forming, (R)- and (S)-acetoin-specific (EC 1.1.1.4), and acetolactate synthase large subunit (EC 2.2.1.6). Acetoin found in *K. marxianus* genome strains is known to be produced during the alcohol fermentation process [45], especially by yeast, including *Saccharomyces* yeasts and the yeasts of other genera *Zygosaccharomyces*, qand can be considered as a precursor for the biosynthesis of 2, 3 butanediols essential for wine production [46]. The encoded protein that is responsible for the biosynthesis of acetolactate synthase subunits is acetolactate synthase large subunit (EC 2.2.1.6), which is also the protein classified as one of those responsible for acetoin and butanediol metabolism. L-lactate dehydrogenase (EC 1.1.1.27) was found responsible for lactic acid fermentation in *K. marxianus* genome strains, an enzyme involved in with lactate oxidation is present in yeast, and its expression can yield the production of lactic acid which is economically important organic acid with a wide range of industrial uses such as food and preservative pH buffering agent, preservative and flavoring compound [47]. We evaluated the *K. marxianus* DMKU3-1042 and discovered that the strain has the encoded proteins of acetoin and butanediol metabolism, acetolactate synthase subunits, and lactate fermentation of lactate (five, two, and three, respectively) associating to the fermentation subsystem names as earlier highlighted. This insight is also common across other investigated strains including SLP1 000000F_001 to have six encoded proteins for Acetoin and butanediol metabolism and in Olga-2 strain (5), for Acetolactate synthase subunits only four proteins were encoded, but three in Olga-2. For fermentation of lactate, we found three encoded proteins responsible and two of them in Olga-2, while the rest of the strains shared the same copies of encoded proteins as those of the *K. marxianus* DMKU3-1042 isolates.

### 3.7. Vitamins, Coenzymes, and Prosthetic Groups

Considering vitamins, coenzymes, and prosthetic groups as part of the metabolism functional categories, we compared the *K. marxianus* strains based on the aggregate number of encoded gene contents found in each subsystem name. In our analysis, we discovered a close range in numbers of the encoded genes when compared to other investigated strains for each of the subsystem names as shown in Figure 5. Vitamins are essentially needed by the natural host due to the nutritional benefits required to perform biochemical reactions by all living cells [48] and in relation to this category, we found six main subsystem names (thiamin, riboflavin, pyridoxine, biotin, folate, lipoic acid) in all the genome strains with a total of 27 encoding genes. Specifically, from our isolates (DMKU3-1042_S1, DMKU3-1042_S2, DMKU3-1042_S3, DMKU3-1042_S4, and DMKU3-1042_S5), the encoded genes were found to play important role in the biosynthesis of various vitamins including B1 (Thiamin), B2 (Riboflavin), B6 (Pyridoxine), B7 (Biotin), B9 (Folate), and B-complex vitamin (lipoic acid). Thiamin is encoded by a gene (NUDIX hydrolase, associated with Thiamin pyrophosphokinase) and was found common across the compared strains. Coenzyme as a key component that is involved in cellular metabolism, notable for its role in a large portion of the cellular oxidative pathways including carbohydrate and amino acid oxidations, fatty acid β-oxidation, and the Krebs cycle [49]. Considering this, we found ranges of encoding genes for 2 Coenzyme A subsystem names (Coenzyme A—gjo and Coenzyme A biosynthesis cluster) in our isolates, as well as other investigated strains (Figure 5). Based on the Coenzyme A—gjo, the genes revealed in all strains include two copies encoding phosphopantothenoylcysteine decarboxylase (EC 4.1.1.36), pantoate–beta-alanine ligase (EC 6.3.2.1), and phosphopantothenoylcysteine synthetase (EC 6.3.2.5) while in Coenzyme A biosynthesis cluster includes two copies encoding 2-dehydropantoate 2-reductase (EC 1.1.1.169), 3-methyl-2-oxobutanoate hydroxymethyltransferase (EC 2.1.2.11), and pantoate–beta-alanine ligase (EC 6.3.2.1). Other subsystems known as prosthetic groups in this category include heme biosynthesis, lipoylated proteins, and metal chelatases.

### 3.8. Stress Response and Defense Mechanisms

*K. marxianus* is known to possess survival response, especially during production on an industrial scale and also at storage processes, it can tolerate the gastrointestinal tract acid environment due to the host’s activity of gastric juice (stomach pH) [50,51]. These survival rates showed the potential of the studied strains ability to respond to elevated temperatures via heat-shock response (HSR) [52]. Specifically, the tolerance to temperature as a means for survival was genotypically evaluated in the isolates of *K. marxianus* DMKU3-1042 genomes. We found encoding genes that play an essential role in heat and hyperosmotic shock tolerance, including GrpE, GroES, GroEl, and DnaK. These genes are classified as heat-shock proteins and function as molecular chaperones performed by folding proteins to carry out a vital role in various cellular functions. There is always a reaction between DnaK, GroES, GroEl, and GrpE in yeast, which usually leads to a sequential interaction during the folding polypeptide [53]. Hundreds of transcribed genes are usually upregulated at increasing temperatures, but the reactions and molecular chaperones in *K. marxianus* will enable the yeast to efficiently buffer phototoxic stress [54]. These findings are relatively common to other investigated strains.

Reactive oxygen species are highly reactive chemicals regarded as toxic byproducts known to cause serious damage to DNA, RNA, and protein and may eventually cause death to the organism but to ensure survival the organism must adequately deploy antioxidant machinery for protection [55]. In *K. marxianus* DMKU3-1042, we found encoding genes relevant for protection from reactive oxygen species including SOD2, Cu-Zn SOD, and KatE encoding superoxide dismutase (Mn), superoxide dismutase (Cu-Zn) precursor [EC 1.15.1.1], and catalase KatE [EC 1. 11. 1.6], respectively. These genes range deferentially across the investigated strains. Glutathione is another important substance that is associated with the ability to protect the organism from reactive oxygen species [56]. It prevents damage to vital cellular components caused by reactive oxygen species and other sources, including heavy metals and free radicals [57]. Across the *K. marxianus* strains in this study, we found three forms of glutathione: the redox cycle, non-redox reaction, and biosynthesis, as well as the gamma–glutamyl cycle (Figure 6); each of these forms contains putative functional proteins which occurred differently when compared between strains of *K. marxianus.* The form with the most abundant encoded functional proteins is those that are associated with biosynthesis and gamma–glutamyl cycle including 5-oxoprolinase (EC 3.5.2.9), HyuA like domain/5-oxoprolinase (EC 3.5.2.9) and HyuB-like domain which was evaluated as the most dominant even across strains, glutathione synthetase (EC 6.3.2.3), gamma-glutamyl transpeptidase (EC 2.3.2.2), and glutathione hydrolase (EC 3.4.19.13).

### 3.9. Amino Acids

Amino acids are known to be the major building blocks of all cells, as well as substrates for the synthesis of low molecular weight substances, essential for their growth and survival [58]. Some amino acids are considered important metabolites, and they are essentially for animals’ nutrition. Yeast utilizes most of the proteinogenic amino acids as a source of nitrogen, hence amino acids are not only necessarily required by the *K. marxianus* strains for rapid growth rate but also for nutritional purposes such as dietary formulations, lactation, growth, etc. [59]. In *K. marxianus* DMKU3-1042 isolates, we evaluated the amino acids present and their putative functions as shown in Appendix A. We showed the presence of amino acids anabolic enzyme genes across the isolates, and relative to the investigated strains. The encoded proteins necessary for the synthesis of lysine, threonine, methionine, and cysteine across all genome strains are homoserine kinase, homoserine dehydrogenase, aspartokinase, Aspartate-semialdehyde dehydrogenase, threonine synthase, S-methyl-5-thioribose-1-phosphate isomerase, homocysteine S-methyltransferase, 2,3-diketo-5-methylthiopentyl-1-phosphate enolase-phosphatase, methylthioribulose-1-phosphate dehydratase,5’-methylthioadenosine phosphorylase, and 1,2-dihydroxy-3-keto-5-methylthiopentene dioxygenase. When compared across the investigated strains, there is a close relativity of the encoded proteins. Imidazoleglycerol-phosphate dehydratase, histidinol-phosphate aminotransferase, imidazole glycerol phosphate synthase, adenylosuccinate synthetase, and phosphoribosylformimino-5-aminoimidazole carboxamide ribotide isomerase are all contained variably across isolates necessary for the synthesis of histidine metabolism. The aromatic amino acids are another interesting one when compared across strains and the encoded proteins include prephenate dehydratase, chorismate mutase III, prephenate and/or arogenate dehydrogenase, phosphoribosylanthranilate isomerase, tryptophan synthase beta chain, and anthranilate synthase. Two encoded proteins (phosphoribosyl anthranilate isomerase and tryptophan synthase beta chain) were detected in DMKU3-1042 isolates. The amino acids proline and 4-hydrxyproline are synthesized by nine encoded proteins (Gamma-glutamyl phosphate reductase, glutamate-5-kinase, pyrroline-5-carboxylate reductase, NADP-specific glutamate dehydrogenase, delta-1-pyrroline-5-carboxylate dehydrogenase, Argininosuccinate synthase, argininosuccinate lyase, ornithine aminotransferase, n-acetylornithine aminotransferase, and Arginase). Alanine, serine, and glycine amino acids synthesis are encoded by phosphoserine aminotransferase, aminomethyl transferase, glycine dehydrogenase, and glycine cleavage system H protein. They are widely common across all strains of *K. marxianus* used in this study, including our isolates. As for the encoded proteins that were associated to the synthesis of glutamine, glutamate, aspartate, asparagine, and ammonia amino acids, they were all detected in across the genome of *K. marxianus* DMKU3-1042 isolates and likely distributed across all investigated strains and likewise the arginine, urea cycle, creatine, and polyamines amino acids.

### 3.10. Antioxidants, Anticoagulants, and Anti-Inflammatory Related Genes

Antioxidants are essential substances considered nutritionally valuable to humans and higher animals and they can be synthesized as bioactive compounds by yeast [60]. In *K. marxianus* DMKU3-1042 genome isolates, at the metabolism subsystem category as shown in Figure 3, we found relevant compounds including fatty acids, lipids, and isoprenoids with encoded proteins responsible for the significant activities of antioxidants and likewise anticoagulants, and anti-inflammatory activities. The fatty acids consist of Acyl carrier protein and Enoyl-[ACP] reductases disambiguation. The proteins associated with the biosynthesis of fatty acids in the formation of Enoyl-[ACP] reductases disambiguation include [Acyl-carrier-protein] acetyltransferase of FASI (EC 2.3.1.38)/Enoyl-[acyl-carrier-protein] reductase [NADPH, Si-specific] (EC 1.3.1.10)/3-hydroxypalmitoyl-[acyl-carrier-protein] dehydratase of FASI (EC 4.2.1.61)/[Acyl-carrier-protein] malonyl transferase of FASI (EC 2.3.1.39)/[Acyl-carrier-protein] palmitoyl transferase of FASI (EC 2.3.1.-). In our analysis, we found two forms of lipids, which include phospholipids and sphingolipids; both were associated with antioxidant and anti-inflammatory roles. In the phospholipids found in the *K. marxianus* DMKU3-1042 genome isolates, one–two copies of CDP-diacylglycerol--glycerol-3-phosphate 3-phosphatidyltransferase (EC 2.7.8.5) functional protein encoding the biosynthesis of cardiolipin were present, while functional protein encoding sphingolipids include serine palmitoyl transferase, subunit LCB1 (EC 2.3.1.50), serine palmitoyl transferase, subunit LCB2 (EC 2.3.1.50), acyl-CoA-dependent ceramide synthase (EC 2.3.1.24), sphingoid long chain base kinase (EC 2.7.1.91), and cytochrome b5 domain/Sphingolipid (R)-alpha-hydroxylase FAH1 (no EC). These proteins were detected across *K. marxianus* with different genes copies number. Yeasts serve as an ideal host for isoprenoid and other terpene based compound synthesis through biotransformation approach [61]. They are of valuable commercial interest with uses as *antioxidants**,*** pharmaceuticals, cosmetics, food additives, and potential advanced biofuels and are also known to exert anti-inflammatory activities [62]. We found isoprenoid biosynthesis (interconversions) and mevalonate metabolic pathway in our analysis. In yeast, the biosynthesis pathways were constructed and optimized by certain metabolic factors including the activity of cofactors and the mevalonate metabolic pathway [63]. The encoded protein essential for isoprenoid biosynthesis (interconversions) were isopentenyl-diphosphate delta-isomerase (EC 5.3.3.2) and farnesyl pyrophosphate synthetase (FPP synthetase) (FPS) (Farnesyl diphosphate synthetase)/dimethylallyltransferase (EC 2.5.1.1)/(2E, 6E)-farnesyl diphosphate synthase (EC 2.5.1.10). Meanwhile, the five isolates of *K. marxianus* strain DMKU3-1042 possess these encoded proteins: mevalonate kinase (EC 2.7.1.36), diphosphomevalonate decarboxylase (EC 4.1.1.33), hydroxymethylglutaryl-CoA synthase (EC 2.3.3.10), and hydroxymethylglutaryl-CoA reductase (EC 1.1.1.34). These were encoded and are essential for the mevalonate metabolic pathway in *K. marxianus* strains and were observed among the studied strain DMKU3-1042 isolates. Other encoded proteins with anti-inflammatory activities were those in reactive oxygen species as earlier discussed including glutathione.

## 4. Conclusions

*Kluyveromyces marxianus*, a yeast that has been predominantly studied for its thermotolerance ability and ethanol production, exhibits considerable potential as a probiotic agent. A comprehensive investigation of the genome was conducted to verify and authenticate the probiotic capabilities of the strain *K. marxianus* DMKU3-1042 isolates from yogurt samples. Furthermore, a comparison was made between the genomes of physiologically and geographically distinct strains from the NCBI dataset. The GC content of the investigated genomes ranged from 40.10 to 40.60%, with projected coding sequences ranging from 9870–9875. Upon conducting a functional genome analysis, it was determined that the isolates possessed genes for sugar utilization, fermentative ethanol production, vitamin and coenzyme synthesis, and amino acid synthesis, as well as anticoagulation and anti-inflammation. Upon comparative genome analysis, difference between the types and number of probiotic genes were observed. In conclusion, we have provided a snapshot of the probiotic potential of *K. marxianus* DMKU3-1042. However, in vitro studies are necessary to further authenticate and explore the probiotic potential of this strain.

## Figures and Tables

**Figure 1 foods-12-04329-f001:**
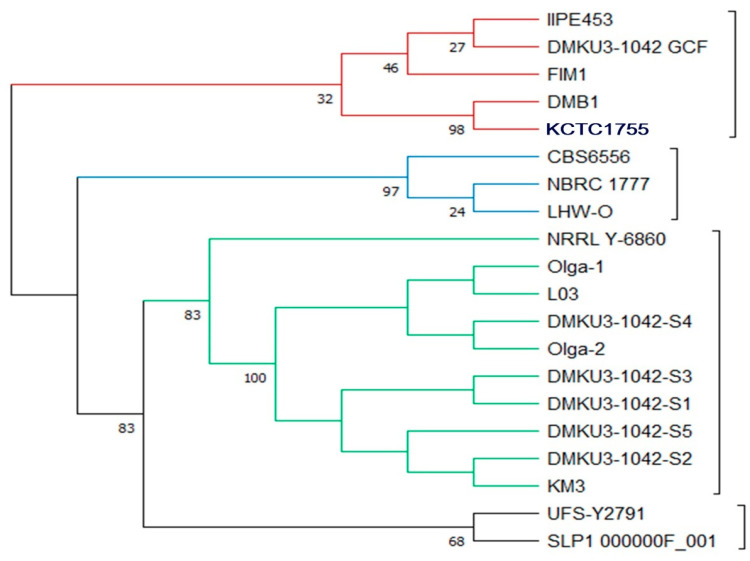
Whole-genome-based phylogenetic tree of 5 isolates of *Kluyveromyces marxianus* DMKU3-1042 genomes compared to the genomes of other strains. To generate the support values of the tree, 100 rounds of RAxML Fast Bootstrapping option were used.

**Figure 2 foods-12-04329-f002:**
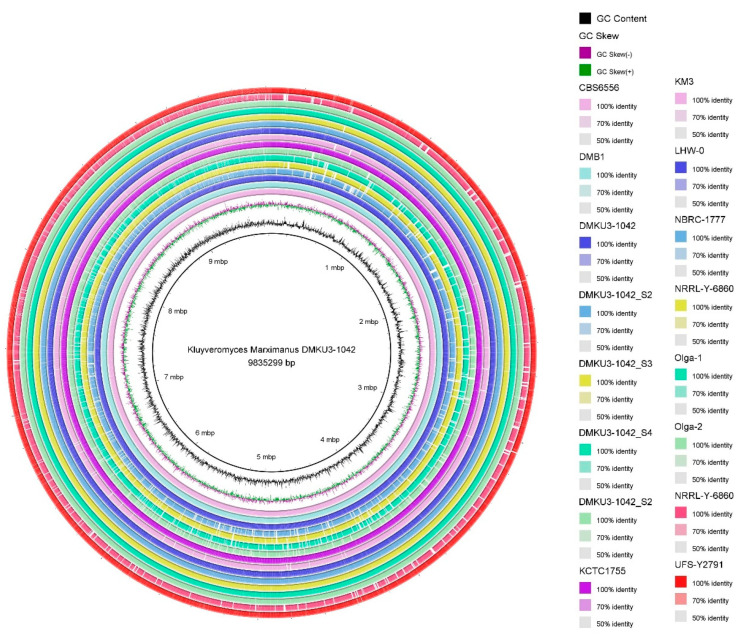
Whole-genome comparison of *Kluyveromyces marxianus* DMKU3-1042 genomes with other yeast strains.

**Figure 3 foods-12-04329-f003:**
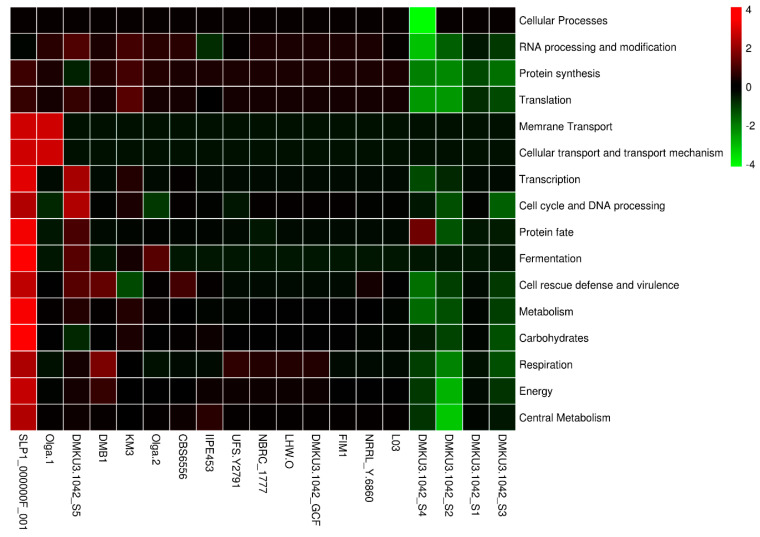
Functional categorization of all predicted coding sequences (CDS) in *Kluyveromyces marxianus* strains.

**Figure 4 foods-12-04329-f004:**
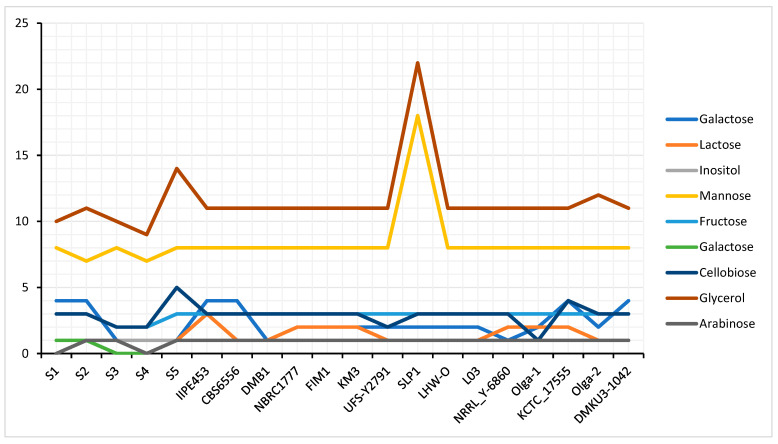
Relative abundance of sugars between *K. marxianus* strain DMKU3-1042 isolates (S1, S2, S3, S4, S5) and other investigated strains in this study.

**Figure 5 foods-12-04329-f005:**
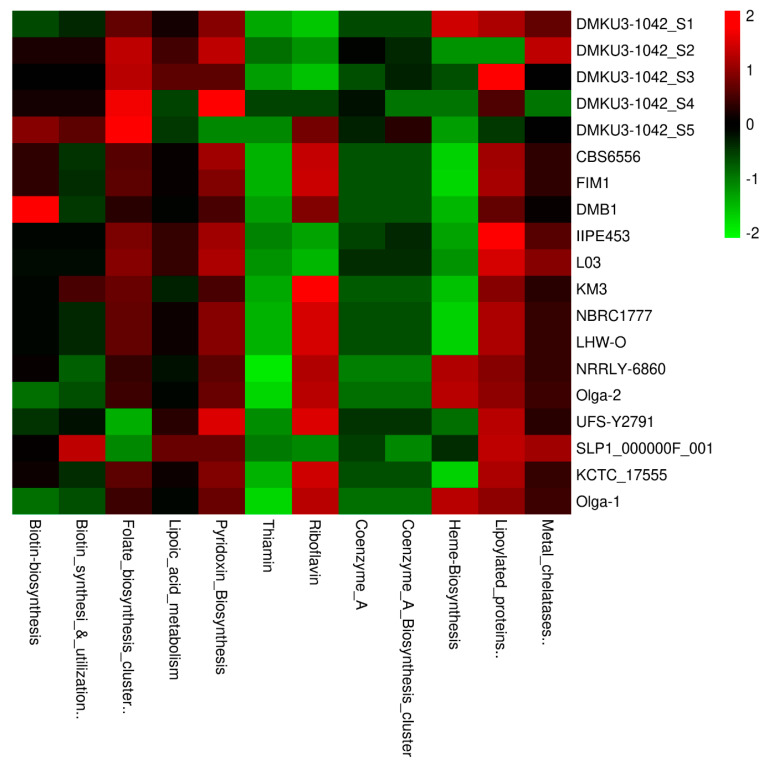
Relative abundance of vitamins, coenzymes, and prosthetic groups across *K. marxianus* strains in this study.

**Figure 6 foods-12-04329-f006:**
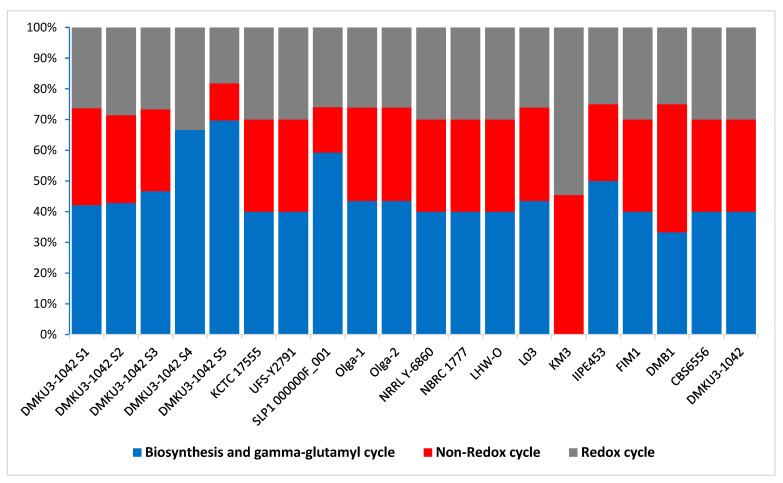
The percentage frequency of the three forms of glutathione across *K. marxianus* strains.

**Table 1 foods-12-04329-t001:** Chromosomal Properties of *Kluyveromyces marxianus* strains.

Species	Genome Size (bp)	Average G + C Content (%)	Total CDS	Total tRNA Genes	Average Gene Density (%)	Average G + C in CDS (%)
DMKU3-1042_S1	9,835,299	40.59	9875	124	71.25	41.69
DMKU3-1042_S2	9,889,048	40.18	9870	121	71.60	41.10
DMKU3-1042_S3	9,806,639	40.20	9870	123	71.20	41.60
DMKU3-1042_S4	9,850,219	40.10	9875	128	71.10	41.25
DMKU3-1042_S5	9,843,168	40.2	9870	134	71.3	41.9
NRRL Y-6860	10,837,618	40.2	8730	170	-	-
Olga-1	10,595,223	40.13	10,209	94	-	-
Olga-2	10,651,240	40.19	10,489	121	-	-
SLP1000000F_001	9,555,365	39.91	20,917	157	-	-
UFS-Y2791	10,695,463	40.4	10,446	26	-	-
KM3	10,592,351	40.21	9791	167	-	-
L03	10,366,177	40.13	9368	11	88.98	46.74
LHW-O	10,776,015	40.15	10,225	180	87.6	46.81
NBRC 1777	10,895,581	40.12	8847	182	-	-
DMKU3-1042	10,966,467	40.12	8821	177	87.36	46.79
KCTC1755	10,920,632	40.14	8890	178	87.31	40.61
CBS6556	10,894,425	40.25	8773	169	87.2	46.64
FIM1	10,914,453	40.16	9167	172	87.96	46.78
IIPE453	10,711,999	40.17	9228	145	93.23	46.68
DMB1	11,165,408	40.09	9384	197	86.79	46.75

**Table 2 foods-12-04329-t002:** Whole-genome-based average nucleotide identity (ANI) of *Kluyveromyces marxianus* genomes compared to the genomes of other investigated strains.

DMKU3-1042 S1																			
99.66	DMKU3-1042 S2																		
99.72	99.77	DMKU3-1042 S3																	
99.67	99.74	99.71	DMKU3-1042 S4																
99.37	99.39	99.36	99.38	DMKU3-1042 S5															
97.16	97.13	97.05	97.12	97.46	CBS6556														
97.19	97.16	97.09	97.16	97.54	98.88	DMB1													
97.21	97.18	97.12	97.16	97.53	98.99	99.08	FIM1												
97.19	97.13	97.07	97.14	97.52	98.96	99.24	99.18	IIPE453											
97.17	97.14	97.07	97.13	97.48	99.86	98.87	98.98	99.04	KCTC1755										
98.48	98.50	98.45	98.47	98.28	97.71	97.71	97.79	97.82	97.75	KM3									
97.23	97.22	97.13	97.20	97.54	97.19	99.04	99.13	99.12	99.23	97.88	LHW-O								
99.55	99.56	99.54	99.57	98.89	97.00	96.97	97.04	97.06	96.98	98.44	97.07	LO3							
97.81	97.16	97.07	97.15	97.52	99.03	99.10	99.07	99.23	99.09	97.81	99.26	97.15	NBRC1777						
97.57	97.56	97.50	97.57	97.59	98.15	98.21	98.22	98.32	98.19	97.66	98.36	97.57	98.19	NRRLY-6880					
99.34	99.31	99.32	99.32	98.72	97.02	97.02	97.09	97.13	97.04	98.26	97.15	99.42	97.03	97.46	Olga1				
99.32	99.31	99.29	99.31	98.70	97.03	97.00	97.07	97.12	97.04	98.26	97.13	99.41	97.03	97.45	99.82	Olga2			
97.16	97.15	97.07	97.14	97.52	98.97	99.25	99.13	99.37	99.03	97.79	99.15	97.15	99.12	98.29	97.10	97.10	DMKU3-1042_GCF		
95.25	95.15	95.08	95.17	95.27	95.87	95.90	95.94	95.92	95.93	95.27	96.03	95.19	95.91	95.44	95.16	95.15	95.85	SLP1	
94.63	94.60	94.49	94.57	94.33	94.50	94.40	94.48	94.62	94.51	94.46	94.60	94.57	94.47	94.48	94.55	94.56	94.46	95.85	UFS-Y2791

## Data Availability

The sequencing raw data used in this study have been deposited in NCBI and allocated the accession number of PRJNA942237.

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
