# Peer review of "Snapshot of the Probiotic Potential of Kluveromyces marxianus DMKU-1042 Using a Comparative Probiogenomics Approach"

_foods, 2023, doi:10.3390/foods12234329_

Round 1

Reviewer 1 Report

Comments and Suggestions for Authors

Comments

1.     Abstract can be more constructive.

2.     At page 1, Line no.30-33, should be rewrite for better clarity.

3.     At page 2, line no. 53, “Protein phosphatases” should be written as “protein phosphatases”

4.     At page 2, line no. 41-42, “Numerous microorganisms predominantly bacteria, and yeast are reported with enormous potential of biomolecules production” author should write name of few bacteria and yeast that have been used for producing biomolecules with appropriate citations.

5.     At page 3,line no. 96-98” Physiological traits such as meta- 96 bolic profile, enzyme and vitamin production, antibiotic resistance, and antipathogenic 97 activity, are determined by genetic material of probiotics” author should cite appropriate reference.

6.     At page 3,line no. 100, full form of any abbreviation should be used first  like

Full form of  “WHO” should be incorporated first.

7.     Typological errors should be corrected before submitting revision.

Specific comments

1.     In abstract, line no. 26, examined strain isolates??

2.     In Materials and Methods, line no. 112, what is pH of YPD broth??

3.     Page 4, line no. 126, NCBI SRA??

4.     Author should mention the novelty of this work, how it will be novel than previous research work reported by other researchers.

5.     Page 5, In results and discussion section, section 3.1, 3.2, 3.3 are missing discussion and not cited any references. Author should check and incorporate discuss appropriate studies in these sections.

6.     Page 19, heading 6. Patents ??please check this.

Comments on the Quality of English Language

Ms. should be checked by an English expert

Author Response

Dear Sir/Madam,

We appreciate the time and effort that you dedicated to providing feedback on our manuscript and are grateful for the insightful comments on and valuable improvements to our paper. We have incorporated most of the suggestions made by you. Those changes are highlighted in yellow within the revised manuscript. Please find attached the point-by-point response file to your comments. Thank you.

Reviewer 2 Report

Comments and Suggestions for Authors

This contribution is interesting and may be published in Foods. Subject of this manuscript is in good correlation with the journal. However, there are mistakes which have to be corrected.

 Comments:

- the abstract part should be a little shorten

- in the introduction part should be highlighted the main aim of the paper, and additionally, what is the novelty of carried research work

- references should be corrected in accordance with the journal's requirements

- the english should be checked

Comments on the Quality of English Language

The english should be checked

Author Response

Dear Sir/Madam,

We appreciate the time and effort that you dedicated to providing feedback on our manuscript and are grateful for the insightful comments on and valuable improvements to our paper. We have incorporated most of the suggestions made by you. Those changes are highlighted in yellow within the revised manuscript.  

Reviewer 3 Report

Comments and Suggestions for Authors

The detailed information in methods and materials section should be added. In the method section, need to be revised due to lack of data; standard laboratory chemicals and equipment, and etc.

In the materials sections, supplier and manufacturer location (city, state where applicable, country) must be included within parentheses after the first citation of a given supplier/manufacturer.

In the methods sections,, explain how yeast isolation and identification of K. marxianus DMKU 3-1042 of fermented traditional yogurt. For example, explain about number of sample, yogurt original source (city and country), and isolation sources.

Please, the use full word in the all manuscript for example, YDP after the first full name.

In the text, you write: Yeast cultures were routinely cultivated in YPD broth (1% yeast extract, 2% peptone, 2% glucose) at temperature of 30°C. Please explain, time of incubation and other conditions, clearly.

At the end, it is necessary to state that in vitro studies are necessary for the probiotic potential of this strain.

Author Response

Dear Sir/Madam,

We appreciate the time and effort that you dedicated to providing feedback on our manuscript and are grateful for the insightful comments and valuable improvements to our paper. We have incorporated most of the suggestions made by you. Those changes are highlighted in yellow within the revised manuscript.
